# Reduced Potential Barrier of Sodium-Substituted Disordered Rocksalt Cathode for Oxygen Evolution Electrocatalysts

**DOI:** 10.3390/nano13010010

**Published:** 2022-12-20

**Authors:** Aditya Narayan Singh, Amir Hajibabaei, Miran Ha, Abhishek Meena, Hyun-Seok Kim, Chinna Bathula, Kyung-Wan Nam

**Affiliations:** 1Department of Energy and Materials Engineering, Dongguk University—Seoul, Seoul 04620, Republic of Korea; 2Center for Superfunctional Materials, Department of Chemistry, Ulsan National Institute of Science and Technology (UNIST), 50, UNIST-gil, Ulsan 44919, Republic of Korea; 3Division of Physics and Semiconductor Science, Dongguk University—Seoul, Seoul 04620, Republic of Korea; 4Division of Electronics and Electrical Engineering, Dongguk University—Seoul, Seoul 04620, Republic of Korea; 5Center for Next Generation Energy and Electronic Materials, Dongguk University—Seoul, Seoul 04620, Republic of Korea

**Keywords:** cation disordered rocksalt, oxygen evolution reaction, sparse Gaussian process potential, machine learning, density functional theory

## Abstract

Cation-disordered rocksalt (DRX) cathodes have been viewed as next-generation high-energy density materials surpassing conventional layered cathodes for lithium-ion battery (LIB) technology. Utilizing the opportunity of a better cation mixing facility in DRX, we synthesize Na-doped DRX as an efficient electrocatalyst toward oxygen evolution reaction (OER). This novel OER electrocatalyst generates a current density of 10 mA cm^−2^ at an overpotential (η) of 270 mV, Tafel slope of 67.5 mV dec^−1^, and long-term stability >5.5 days’ superior to benchmark IrO_2_ (η = 330 mV with Tafel slope = 74.8 mV dec^−1^). This superior electrochemical behavior is well supported by experiment and sparse Gaussian process potential (SGPP) machine learning-based search for minimum energy structure. Moreover, as oxygen binding energy (O_BE_) on the surface closely relates to OER activity, our density functional theory (DFT) calculations reveal that Na-doping assists in facile O_2_ evolution (O_BE_ = 5.45 eV) compared with pristine-DRX (6.51 eV).

## 1. Introduction

As our reliance on eco-friendly and renewable energy grows, there is an urgent need to explore various sustainable sources. In this race, solar energy is considered an enormous energy source that can permanently solve the world’s energy crisis [1,2]. However, a few hours of sunshine in a day, uneven power density distribution on the earth due to geographical area, and low power conversion efficiencies of photovoltaic devices threaten its widespread acceptability [3,4]. Of several other renewable energy resources, the splitting of water to generate hydrogen as a fuel and oxygen as a byproduct is the most viable technique. However, in several electrochemical reactions, the energy conversion rate and efficiency are often limited by oxygen evolution reaction (OER) expressed as Equation (1) in aqueous media [5]. This reaction proceeds via multiple-state reactions involving 4 electron oxidations, thus, becoming kinetically sluggish. The η needs to be lowered to accelerate this reaction kinetically. Therefore, several OER electrocatalysts exhibiting lower η have been explored so far [6,7,8].
(1)$$4OH^{-}\to 2H_{2}O+4e^{-}+O_{2}\;;\;E=1.23\;V\;versus\;RHE$$

Over the past few decades, numerous 3d transition metal (TM)-based layered oxide cathodes have been explored to act as OER catalysts [6,7,8,9,10]. It has also been reported that the properties of a Co-based, efficient OER electrocatalyst, particularly Co_3_O_4_ [11], can be effectively tuned up by lithium insertion/deinsertion as in LiCoO_2_ [7]. Apart from a single 3d TM containing OER electrocatalysts, a dual 3d TM Fe-substituted LiNiO_2_ has also been reported to be an efficient electrocatalyst [8]. The higher activity of this doubly doped layered oxide is attributed to the higher oxidation states of TMs and their stabilization in layered structures [12]. So far, several layered oxides have only been looked upon as a suitable host for stabilizing the highly oxidized TM species necessary for effective OER catalysis. Though layered oxides are better suited to stabilize oxidized TM species, it has also been predicted that the electronic state regulation is another avenue to enhance the activity of perovskite electrocatalysts [13]. In this realm, doping is an effective strategy to tailor the oxidation states of TM ions in layered oxides, to effectively promote hole hopping between TM^n^ to TM^(n+1)^ ions [14,15]. However, the binding energy (BE) of surface oxygen is another vital aspect in OER catalysis, which has rarely been considered in previous reports. Thus, the study must be taken up to understand the role of BE of TMs with surface oxygen in the presence of doped cations/anions.

Herein we report, for the very first time, a uniquely combined 3d/4d TMs-based cation-disordered rocksalt (DRX) cathode [16] material with Na-substitution in the lattice framework by the facile solid-state route. This uniquely combined 3d/4d TMs DRX cathode is believed to entirely change the stacking sequence to that of just 3d TMs cathode while doping of monovalent cation substitution assists in promoting the oxidation state of TMs. The X-ray diffraction and other characterization techniques reveal that Na-substitution significantly assists in OER. The Na-substituted disordered cathode achieves a small Tafel slope of 67.5 mV dec^−1^, η of 270 mV at a current density of 10 mA cm^−2^, better than that of the IrO_2_ benchmark catalyst, and long-term stability over 133 h. The machine learning aid to search for the minimum energy structure along with density functional theory (DFT) calculations reveals that the oxygen BE is ~6.51 eV in pristine. In comparison, it is much lower (~5.45 eV) in Na-substituted cathode. The higher activity in the Na-substituted DRX cathode is attributed to the facile desorption of O_2_ from the surface (Figure 1).

## 2. Experimental Section

### 2.1. Materials and Reagents

Lithium nitrate (7790-69-4) and Ru (IV) oxide anhydrous (12036-10-1) were purchased from Acros Organics (Fair Lawn, NJ, USA). Sodium nitrate (7631-99-4) and nickel acetate tetrahydrate (6018-89-9) were purchased from Sigma-Aldrich (St. Louis, MO, USA). Acetone was purchased from Samchun Chemicals (Gangnam-gu, Seoul, Republic of Korea).

### 2.2. Syntheses of Disordered Cathode Active Material

We opted for a conventional solid-state reaction method with a slight modification to synthesize cathode active materials calcined at 950 °C. DRX cathode materials, precisely Li_1.22_Ru_0.61_Ni_0.16_O_2_ (**1**) and Li_1.22_Ru_0.61_Na_0.05_Ni_0.10_O_2_ (**2**) were synthesized. The powders of LiNO_3_, NaNO_3_, Ni(CH_3_COO)_2_.4H_2_O, and RuO_2_ were stoichiometrically calculated and thoroughly homogenized using mortar and pestle for around 10 min. Subsequently, the powders were mixed with acetone, followed by ball milling for the next four hours. To compensate the loss during high-temperature synthesis, an excess of 5 wt% of LiNO_3_ and NaNO_3_ was taken. The homogenized mixture was then dried overnight in a vacuum oven at 100 °C. The powders were again mortar-pestled, and finally, powders were transferred in an alumina crucible for sintering at 950 °C for 12 h at a ramp rate of 5 °C/min in a tube furnace in the ambient atmosphere. Subsequently, the furnace was cooled to room temperature, and samples were ground manually to obtain a fine powder.

### 2.3. Materials Characterization

To identify the crystal structures, high-power powder X-ray diffraction (HP-PXRD) data were obtained on a Rigaku X-ray diffractometer (3 phase, 380 V, 18 kW) equipped with Cu Kα radiation (λ = 1.54 Å) in the 2θ range of 10–80°. The XRD samples were prepared by depositing a thick film of cathode powder on a glass substrate. To reveal the morphology and chemical composition of samples, a JEM-2200FS (Cs corrected STEM) HRTEM coupled with an energy-dispersive X-ray spectrometer (EDX, Oxford-INCA) at an acceleration voltage of 200 kV was used. Samples for the HRTEM analysis were prepared following a conventional standard procedure by dispersing in ethanol and then sonicating for a while. The well-dispersed suspension was dropped on TEM Cu-grid and oven-dried to use for HRTEM analysis.

### 2.4. ICP-OES

Inductively coupled plasma optical emission spectrometry (ICP-OES) was used to determine the exact stoichiometry of the synthesized cathodes.

### 2.5. XPS Data Collection and Analysis

X-ray photoelectron spectroscopy (XPS) data were obtained on K-alpha (thermo fisher, Brighton, UK). The XPS data were analyzed using the commonly known XPS peak fitting software CasaXPS version 2.3.22.

### 2.6. OER

The electrochemical characterizations involved in our study were carried out with a three-electrode system on VSP (BioLogic Science Instruments, Inc., Seyssinet-Pariset, France) with a graphitic rod and a calibrated Hg/HgO (Appendix A) as a counter and reference electrode, respectively. Nickel foam (NF) was engaged as a working electrode. Through a drop-casting method on NF, a 1 mg cm^−2^ loading amount of the catalysts was achieved for all working electrodes. To ensure that experiments are conducted under equilibrium conditions of H_2_O/O_2_, the electrolyte (1M KOH) was continuously kept in flowing oxygen environments for at least 20 min. The linear sweep voltammograms test was performed at a scan rate of 2 mV s^−1^ with 95% iR compensation. Chronopotentiometry at fix current density of 10 mA cm^−2^ is used to evaluate the long-term stability of the catalyst. The measured voltage values were converted to the reversible hydrogen electrode (R.H.E.) using the following equation: $E_{RHE}$ = $E_{\mathrm{Hg}/\mathrm{HgO}}$ + $E_{\mathrm{Hg}/\mathrm{HgO}}^{0}$ + 0.059 × pH. Tafel slopes were obtained from η = b log j + C; (where η, b, j, and C represent overpotential, Tafel slope, current density, and intercept, respectively). Finally, the overpotential value was obtained by η = E vs (R.H.E) − 1.23.

### 2.7. DFT Calculations

The Vienna Ab initio Simulation Package (VASP) [17], which implements the projector augmented-wave [18] approach to DFT with PBE GGA functionals [19], is used for all FP calculations. Calculations are spin-polarized, and a kinetic energy cutoff of 500 eV is applied. Due to the large size of unit cells, the Γ-centered k-Point grid of (1,2,3) is chosen. (U; J) values of (4:5; 0:6) for Ru and (4:7; 1:0) for Ni are utilized for PBE+U calculations. A huge number of potential energy calculations were needed to find the favorable doping positions and the minimum energy structures. A machine learning potential is built on-the-fly for accelerating the structure search with a sparse Gaussian process potential (SGPP) algorithm [20,21,22] as implemented in the AutoForce package.

### 2.8. Genetic Algorithm

After obtaining the unit-cell formula consistent with doping, we searched for the minimum energy structure by switching the relevant atomic positions. A variation of the genetic algorithm (GA) is used. The search starts with a few random parent structures. At each step, tens of children are generated randomly for each parent. Dozens of lowest energy structures among the union of parents and the new generation are kept as parents for the next generation. The search is stopped if the parents set stays mostly the same for a few steps. Several independent investigations are carried out. The search is accelerated with an SGPP machine learning model, which is built on-the-fly. If the ML model is estimated to be inadequate at every energy calculation step, exact DFT calculations are carried out, and the model is updated.

## 3. Results and Discussion

A solid-state synthesis route was used to obtain DRX Li_1.22_Ru_0.61_Ni_0.16_O_2_ and Li_1.22_Ru_0.61_Na_0.05_Ni_0.10_O_2,_ hereafter denoted as **1** and **2**, respectively. The bulk composition was determined using inductively coupled plasma optical emission spectroscopy (ICP-OES), closely following the theoretical composition (Appendix A and Experimental Section). The crystal structure determined using high-power powder X-ray diffraction (HP-PXRD) reveals that the most characteristic XRD peaks (Figure 1a) in both cathodes are assigned to the C2/c space group of Li_2_RuO_3_ [23,24]. The merged peaks ~63° indicate disordered structures of the synthesized cathodes [25]. Compared to **1**, the XRD patterns of **2** show the asymmetric peak broadening effect (marked by the green circle in Figure 1a), a signature of lattice defects/strains/dislocation [26] possibly arising from slightly larger radii of Na^+^ doped in Li slab. These defects reflected in the form of peak broadening in the XRD pattern confirm that Na^+^ is uniformly distributed inside the layered lattice of **2**. The previous report suggests that doping-induced lattice defects are expected to enhance electrochemical performances [27]. To elucidate the electronic structures, the core-level XPS data were obtained. The XPS peak fitting data (Figure 1b and Appendix A) reveal that Na^+^ doping brought significant changes in electronic structure. Before moving to the discussion of XPS peak fitting, we disclose the BE of Na. The BE value of ~1071.38 eV (Figure 1c) corresponds to the Na^+^ states of Na, indicating that Na has been electro-positively doped in the crystal framework. The Ni 2p (Figure 1d and Appendix A) possesses two spin-orbit doublets at 855.28 (Ni 2p_3/2_) and 872.48 eV (Ni 2p_1/2_) accompanied by two satellites at 861.28 and 879.68 eV. The BE ~855.28 eV corresponds to Ni^3+^, whereas deconvolution of Ni 2p_1/2_ reveals two sets of energy bands: 872.48 eV and 873.88 eV for OH^-^/OOH. This XPS revelation of Ni hints toward the presence of more Ni^3+^ species favoring the electrophilicity of adsorbed O, thereby catalyzing the formation of -OOH species from OH^-^ (these steps are considered as rate-determining steps in alkaline media) [4,28]. The electronic structure of Ru in **2** is also different from **1** (Appendix A). The Ru 3d_5/2_ can only be fitted with a doublet (separated by ~1 eV), indicating the presence of both metallic (280.68 eV) and oxidized ruthenium on the surface (Figure 1e). This oxidized ruthenium can be inferred as oxides with low BE (281.98 eV) and with high BE (283.78 eV) [29]. The BEs at 285.38 and 286.68 eV correspond to RuO_x_/Ru. Most importantly, XPS results of the O 1s core-level (Figure 1f), fitted with three components apportioned as oxidative oxygen (O_2_^2−^/O^−^: BE ~529.58 eV), carbonate/hydroxides/oxyhydroxides species (CO_3_^2−^/OH^−^/OOH: BE ~531.48 eV), and absorbed water (H_2_O: BE ~534.88 eV) suggest a relatively more significant number of O_2_^2−^/O− species in **2**. This indicates a surplus amount of surface oxygen vacancies [30] compared to **1** (Appendix A) with hydroxides (BE ~531.7 eV) and oxygen species (BE ~533.4 eV). Firstly, Na doping has significantly modified the electronic structure, which agrees with previous reports on layered cathodes [6,8]. Secondly, the excess of O_2_^2−^/O− species on the surface of **2** is beneficial for catalyzing OER reactions in alkaline media [31,32].

Scanning electron microscopy (SEM) images reveal a distinguishable feature in their morphology (Figure 2a and Appendix A). The particles in **2** show cornered-shaped structures with smaller particles decorating larger particles, thus, creating rougher surfaces. These rough surfaces expose multiple active sites facilitating oxygen evolution reactions [33]. A better sintering ability with enhanced particle connectivity is an additional feature achieved by Na doping. The lattice fringe distance of 0.475 nm revealed under high-resolution transmission electron microscopy (HRTEM) image corresponds to the disordered structure (Figure 2b). As expected, Na^+^ (marked by the pink arrow) distributes uniformly inside the lattice (Figure 2c). Other interesting features that promote OER kinetics can be explored under HRTEM. For instance, defective zones (marked by the yellow arrow) can also be seen in Figure 2c of Na-DRX, whereas no such beneficial features can be seen in **1** (Appendix A**)**. These defective zones are termed as stacking fault defects, primarily arising from the differences in ionic radii of Na^+^ and Li^+^ [34,35]. These defective sites can modulate the electronic structure and tune surface properties of **2**, thus, optimizing the adsorption energies of OER steps [27,36]. The autocorrelated image reveals the Na-doping and defects zone (Figure 2d). The energy-dispersive X-ray spectroscopy (EDS) mapping images show a uniform distribution of elements with a slight increase in elemental O on the surface.

To further corroborate the increased oxygen vacancies through Na-doping, the electrochemical oxygen intercalation in **1** and **2** was probed through cyclic voltammetry (CV) performed on a three-electrode system in oxygen saturated 1M KOH electrolyte environment. As illustrated in Figure 3a, redox peaks appear as oxygen ions are inserted into and extracted from the accessible lattice vacancy sites. It is important to note that **2** shows positive-shifted redox peaks (marked by arrows) compared to **1**, which is in harmony with the increased BE reflected by XPS. Furthermore, the linear sweep voltammetry (LSV) curve of **2** exhibits a low η of 270/390 mV (on NF) to yield current densities of 10/250 mA cm^−2^ (Figure 3b and Appendix A). This catalytic activity is much better than **1** and several other cathode materials deployed for water catalysis (Appendix A). As the reaction kinetics directly relates to the electrochemically active surface area, we also evaluated double-layer capacitance (C_dl_) measurements to compare the electrochemically active surface areas of **1** and **2** (Appendix A). The double-layer capacitance **2** (C_dl_ = 1.63 mF cm^−2^) is higher than **1** (C_dl_ = 1.25 mF cm^−2^), which suggests that the former possesses a relatively higher density of active sites for catalytic reactions, thus, further boosting the OER activities.

To get better insights into OER kinetics, the Tafel slopes of the cathodes were obtained from the steady-state polarization curve. Figure 3c manifests that the Tafel slope of **2** (67.5 mV dec^−1^) is lower than **1** (72.2 mV dec^−1^) and benchmark IrO_2_ (86 mV dec^−1^) [8]. The lower Tafel slope value of **2** indicates that the Na^+^ substitution into the DRX matrix plays a vital role in promoting the electrochemical OER kinetics, thus, improving the intrinsic catalytic activity.

To evaluate the impact of Na^+^ doping on electrical conductivity in **2**, we studied its impedance spectra (Figure 3d and Appendix A). The semicircle is related to the charge-transfer resistance (R_ct_) of the cathode materials. The electrochemical impedance spectroscopy (EIS) spectra show that the charge-transfer resistance of **2** (R_ct_ = 16.01 Ω) is lower than **1** (R_ct_ = 20.89 Ω) and much lower than the commercial IrO_2_ (72.8 Ω), indicating a better electron transfer capability across the electrode/electrolyte interface, thereby indicating a superior OER kinetics and an improved overall electrochemical performance (Figure 3e).

The long-term stability of cathode materials is another crucial factor that deters the wide applicability of DRX materials for water-splitting reactions. We, therefore, investigated the stability in 1M KOH. The chronopotentiometry results suggest that **2** has superb stability of more than 5.5 days and even beyond without an appreciable increment in overpotential (Figure 3f and Appendix A). In contrast, **1** shows a tendency to increase in potential even after a few hours. This superb stability of **2** is attributed to the following factors. The Na doping in the lattice increased the oxidation states of Ru and Ni, thereby modifying the electronic structures. As reported earlier, higher oxidation states of TMs in the layered structure are beneficial for OER [37]. Additionally, these higher oxidation states can be easily stabilized in DRX layered structures, which is essential for sustained OER [38]. Furthermore, it has also been widely accepted that high-valence transition metal cations facilitate the adsorption of OH− with the catalysts to form adsorbed −OOH species to promote OER reaction steps readily [8,39]. The above results reveal the enhanced electrochemical performance of **2** compared to not only **1** but also to many other cathode materials (Figure 3e).

It is generally assumed that the oxygen BE at the surface largely influences the OER activity [8,40,41,42]. For this, we conducted a series of DFT calculations to investigate the role of Na-doping on surface oxygen atoms, as shown in Figure 4. The favorable locations of Ni and Na in **2** are found with Genetics Algorithm (GA). DFT calculations established that while Ni atoms prefer to occupy the Ru layer (Figure 4a), Na atoms prefer to locate in the Li layer (Figure 4b). Surprisingly, the average oxygen BEs in both cases are almost the same (∼7 eV), but while it is almost uniform in **1**, **2** shows higher fluctuations. In particular, the lowest BE for oxygen is 5.45 eV in **2** and 6.51 eV in **1**. The Na-substitution promotes the OER activity by reducing the BE of some oxygen atoms at the surface and accentuating the facile diffusion of O_2_ gas, which is consistent with experimental results. The total density of states (DOS) reveals that the electrons’ occupancy near the Fermi level between **1** and **2** is different. In addition, the projected DOS of oxygen 2p orbital in **2** (Figure 4e) is substantially different from that in **1** (Figure 4d) and shows a shift in the O p-band center toward the Fermi level [43], which leads to better TM-O overlap and creates reduced charge transfer resistance well supported with Figure 3d.

## 4. Summary and Conclusions

In summary, we have enrooted the idea of engaging cation-disordered cathode materials for OER application synthesized by a facile solid-state route. Na-doping in **2** solved the issue of instability prevailing in disordered cathodes but also contradicted the former idea that layered structures alone can stabilize TM oxidized species for improved OER performances. Nonetheless, **2** performs much better than **1** and many other layered cathode materials, including dedicated OER electrocatalysts. **2** requires a low η of 270/390 mV to display current densities of 10/250 mA cm^−2^ and demonstrate superior stability for more than 5.5 days, which is exceptionally better than many of the OER electrocatalysts reported to date. The significant enhancement in the OER performance is attributed to several interesting factors. First, the disordered structure provides better interaction between Li and TMs, often unavailable in layered cathodes where Li and TMs occupy a fixed position. Secondly, the induction of Na^+^ promoted the oxidation state of TMs (Ni and Ru), favoring OER kinetics. Thirdly, DFT calculations predicted that Na^+^ accentuates the facile evolution of O_2_ from the surface layer, thereby alleviating the sluggish reaction kinetics often observed in the OER. In addition, we are in the process of understanding the effect of Na doping to use this material as a battery material, and further works are in the pipeline to understand and explore the possibility of other doping sites (such as Ru or Li) and their influence on the electrochemical performances. We believe that our finding of engaging DRX cathode materials for the OER application will pave a novel design strategy for future materials of this family for high-performance electrocatalysts for energy conversion reactions.

## Data Availability

All the experimental data presented within this article, along with the Appendix A, will be made available from the authors upon reasonable request.

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
