# Peer review of "Reduced Potential Barrier of Sodium-Substituted Disordered Rocksalt Cathode for Oxygen Evolution Electrocatalysts"

_nanomaterials, 2022, doi:10.3390/nano13010010_

Round 1

Reviewer 1 Report

This paper by Nam et al describes the electrocatalytic properties of a newly synthesised sodium-doped disordered rock salt material with improved water-splitting performance.

The paper is well written and organised and reads pleasantly. It presents a thorough study based on complementary experimental techniques and spectroscopies, leading to interesting and relevant conclusions.

As said, I find the paper generally well written and pleasant to read, but the authors do get a little "poetic" here and there in their use of language, making it a bit "over the top" for a scientific communication.

Examples:

"grabbing" in line 19 --> "utilising" better?

"superb" in l 23 and other places --> "superior", "excellent"

"intriguing" in l 313 --> "interesting" or similar

My key scientific concern is with the restriction of the tests to the 5% Na-doped material. Why did the authors pick this specific stoichiometry and, if that's the optimal variant, why didn't they substantiate this with materials of lower/higher doping levels? Also, I do not understand the sentence (l 94, 95) "5% over stoichiometrically calculated amounts for Li and Na. 5% Over what? How to accurately pre-set the composition as required following this procedure?

Minor comments/suggestions

line 50 "lithium insertion/(de)insertion" - replace with either "lithium insertion/deinsertion" or "lithium (de)insertion"

l 62 "efforts ... substantiated" --> " ... results/observations ..."

l 99 "grounded' --> "ground"

124 "carried out"

127 delete "The" at start of sentence

172 "Nadoped in Li slab". Briefly explain

209 "brought" --> use "generated", "created" or similar

212 "exciting" --> use "interesting" or similar instead

239 symbol eta - define on first use

275 - 278 remnant of internal editing process? delete

Legend Fig. 4 - define purple spheres. Shading oxygens unclear/invisible

310 "superb" - see earlier comment

313 "intriguing" - see earlier comment  

Reviewer 2 Report

In this work, Aditya et al. reported ‘Reduced Potential Barrier of Sodium-substituted Disordered Rocksalt Cathode for Oxygen Evolution’ Electrocatalysts in 1.0 M KOH electrolyte with an overpotential (η) of 270 mV at 10 mA cm-2. Overall, the work is interesting, and the experimental and analysis results are informative and convincing. Therefore, I would like to suggest publication after addressing the following issues.

1. The authors should improve all figures/graphs’ quality and must modify the text font size within figures/graphs and color code. In Figure 4 author should provide in detail the color code of Ru, Li, and Ni due to the purple color.

2. The author should provide the active site in your material's DFT calculation to explain why your catalyst performance is improved after doping Na.

3. The authors did not provide the electrochemical impedance spectroscopy (EIS) equivalent circuit for the catalytic systems in Figure 3d.

4. Author should pay attention and carefully check typos errors in the whole manuscript (for example, 2.6OER would be 2.6 OER).

5. The materials should be compared with a commercial catalyst such as RuO2 in both polarized curves and stability tests.

6. Author claimed that the material was very stable in 5.5 days.  The author should provide a structure analysis after the OER test.

7. The authors did not calculate the turnover frequency (TOF) values for Na loading in Li1.22Ru0.61Na0.05Ni0.10O2 and Li1.22Ru0.61Ni0.16O2. Authors are highly encouraged to provide TOF values.

Reviewer 3 Report

Dear authors,

thank you for your work. 

If it’s possible, please, write the announcer for your next research steps of this topic in the end of the conclusion.

Best regards,

Reviewer

Round 2

Reviewer 2 Report

It can be published as it is in this journal